# Quantum and non-local effects offer over 40 dB noise resilience advantage towards quantum lidar

Phillip S. Blakey [1] ✉, Han Liu[1], Georgios Papangelakis[1], Yutian Zhang[1], Zacharie M. Léger [1], Meng Lon Iu[1] & Amr S. Helmy[1]

Non-local effects have the potential to radically move forward quantum enhanced imaging to provide an advantage over classical imaging not only in laboratory environments but practical implementation. In this work, we demonstrate a 43dB higher signal-to-noise ratio (SNR) using a quantum enhanced LiDAR based on time-frequency entanglement compared with a classical phase-insensitive quantum imaging system. Our system can tolerate more than 3 orders of magnitude higher noise than classical single-photon counting quantum imaging systems before detector saturation with a detector dead time of 25ns. To achieve these advantages, we use non-local cancellation of dispersion to take advantage of the strong temporal correlations in photon pairs in spite of the orders of magnitude larger detector temporal uncertainty. We go on to incorporate this scheme with purpose-built scanning collection optics to image non-reflecting targets in an environment with noise.

Quantum photonic technologies have been gaining significant momentum, where generation[1], detection, processing, and utilization of quantum states have been advanced significantly towards practical implementation and integration[2]. This, in turn, offers numerous possibilities for their usage in advancing well-established fields, including sensing, communications, and computing.

In parallel with such advances, several emerging applications including LiDAR[3] and biomedical imaging[4] have been encountering challenges to meet their noise-resilience roadmap goals using practical, robust, and low complexity classical protocols. The recent advances in quantum photonic technologies offer a promising route to addressing many of these challenges.

Quantum Illumination (QI) has been proposed, investigated, and demonstrated over the past decade as a solution to the challenges of combating environmental noise in LiDAR and other imaging applications[5,6]. By noise we refer to the portion of the returned light that is uncorrelated with the presence or absence of the target. When compared with optimal classical detection using a coherent state, QI has been shown to provide the largest improvement theoretically possible[7]. However, to date, implementations of QI have been unable to reach such theoretically predicted bounds irrespective of the approach taken[8]. For both QI and optimal classical coherent detection, the states used must have a stable phase. Many practical limitations make it extremely difficult to keep the interacting waves phase-locked, even with the most complex stabilization systems. As a consequence, optimal coherent states are not usually used in practical classical LiDAR applications, where intensity-based detection schemes are more convenient[9], and therefore using such states as a benchmark may not be representative.

Alternative phase-insensitive quantum enhanced target detection schemes have been reported using quantum correlations, including those in both intensity and time[10,11]. Such phase-insensitive systems provide dramatic simplification in implementation, and are attractive for systems where large phase noise is introduced during operation. This improvement comes at the cost of an increased minimum probability of error at high noise powers when compared with a phase-sensitive approach like Quantum Illumination[12]. Phase-insensitive detection approaches such as refs. 13, 14 have demonstrated significant noise resilience compared with classical phase-insensitive detection; however, an improved performance has been demonstrated over classical phase-insensitive counterparts only in a limited range of noise powers. This has been due to the saturation of detectors

[1]Edward S. Rogers Faculty of Applied Science and Engineering, University of Toronto, Toronto, ON, Canada. ✉e-mail: phil.blakey@mail.utoronto.ca

at high noise levels, where the most significant advantage of such schemes is obtained. Further, the use of strong temporal correlations is limited by the relatively large detector time uncertainty. This uncertainty effectively erases any correlation shorter than the detector time uncertainty, reducing the advantages offered by temporal correlation (as highlighted in Supplementary Note 12). For example, commercial superconducting nanowire single-photon detectors have time uncertainty $\simeq 50$ ps which is several orders of magnitude larger than the shortest achievable correlation times[15]. State-of-the-art detectors have time uncertainty $\simeq 3$ ps[16], however this comes at the cost of reduced efficiency while still being several orders of magnitude larger than the shortest correlation times.

In this work, we utilize quantum temporal correlation to assist in the discrimination of a target from the background noise for LiDAR application. By measuring in a rotated basis between time and frequency—the Fractional Fourier domain—we can magnify the probe-reference temporal uncertainty while maintaining the same degree of correlation. This allows us to fully use the probe-reference correlations to distinguish a target from background noise. The uncorrelated noise is broadened well beyond the detector uncertainty. Applying a suitable temporal window can then filter the noise that no longer overlaps the signal. With this method, we were able to enhance the signal-to-noise ratio by up to 43.1 dB compared with a phase-insensitive classical target detection counterpart using the same probe power. This method retains the ease of implementation of previously mentioned target detection schemes, while also increasing the noise power that can be tolerated before detector saturation.

We then integrate this scheme with a purpose-built telescope to image—in 3D—a target in a highly noisy environment thus demonstrating the applicability of this technique. The telescope allows for a wide scanning angle while maintaining a superior coupling efficiency into a single-mode fiber, when compared to alternative telescope designs.

## Results
### System description
The underlying working principles of our system are illustrated in Fig. 1. First, non-classical temporally correlated photon pairs are generated through femtosecond pumped spontaneous parametric down conversion (SPDC). The pump is chosen to be sufficiently weak such that the expected number of probe and reference photons is much less than one.

The probe photon is sent out into the environment while the reference photon is stored locally. The probe photon incurs loss during propagation towards and returning from the target, reducing the expected photon number in the probe beam (Fig. 1b). During the probe photon's propagation, environmental noise is coupled into the probe path. Working under the assumption that the noise has the same spectral/temporal distribution as the probe photon (Fig. 1c) (as other noise may be filtered out classically), we apply anomalous dispersion to the probe/noise photon. This broadens the temporal distribution of the probe/noise photon, resulting in a lower probability of finding a photon in a finite time window. An equal amount of normal dispersion is then applied to the reference photon which also broadens the temporal distribution of the reference photon (Fig. 1d). A coincidence measurement is then performed on the two paths. Due to the quantum correlations between the probe and reference photons, the effects of the dispersion are canceled and the true coincidence peak appears almost as though the photons were not dispersed[17,18]. Conversely, the noise and reference photons only share classical correlations and so the effects of dispersion cause a broadening of the coincidence peak (Fig. 1f). By choosing an appropriate temporal window, the probability of measuring a false coincidence between a noise photon and a reference photon is reduced, while the probability of measuring a true coincidence between a probe photon and a reference photon is essentially unchanged.

To demonstrate LiDAR capabilities, an apparatus for 3D imaging was designed specifically for use with single-mode fiber (SMF) coupled superconducting nanowire detectors (Fig. 2b). First, the probe photons from the SPDC source are collimated onto a pair of galvanometer mirrors. These mirrors direct the probe photon onto a target in the field of view of a telescope. The rotation of the mirrors allows for scanning of the target in both $x$ and $y$ directions. To reduce the mode mismatch between the collected light and SMF, a negative meniscus lens was used at the telescope aperture to reduce the angle offset.

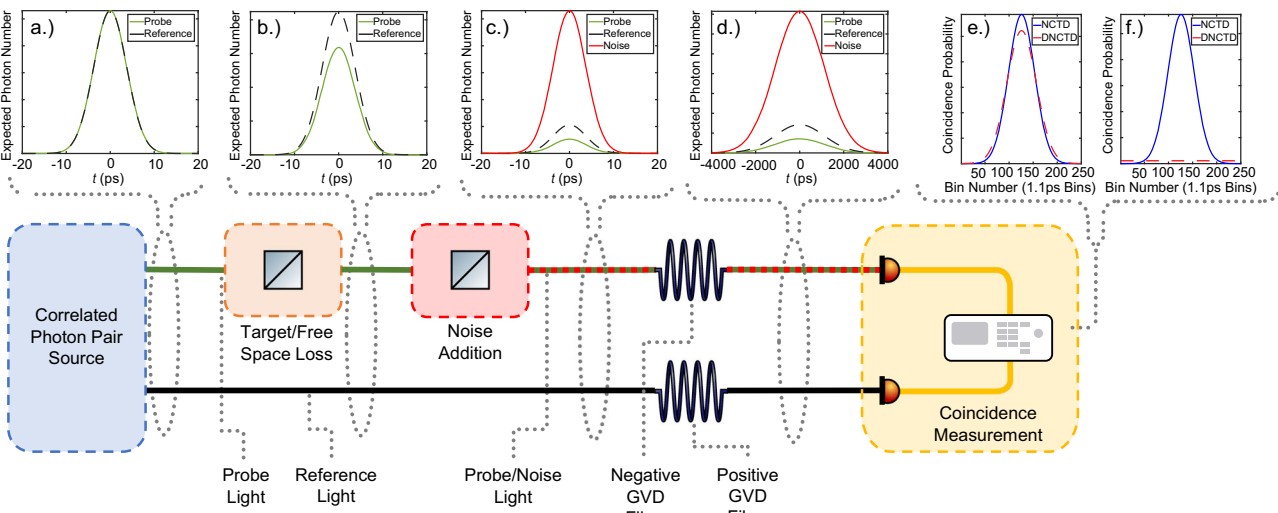

**Fig. 1 | Logical diagram of the experimental setup with illustrative temporal distributions. a** Temporal distribution of probe and reference photons directly after spontaneous parametric down conversion (SPDC). **b** Temporal distribution of the probe and reference photons after free-space/target loss. **c** Temporal distribution of the noise, probe, and reference photons after coupling of environmental noise. **d** Temporal distribution of the noise, probe, and reference photons after positive and negative group-velocity dispersion (GVD) fibers. **f** Theoretical coincidence probability histograms for true coincidences (**e**) and false coincidence (**f**) in both the non-dispersed (blue) and dispersed (red) regimes. The model uses joint-spectral amplitude (JSA) bandwidths of 100 fs and 17.7 ps full-width at half maximum (FWHM) estimated from the Ti:Sapphire laser and second harmonic generation (SHG) spectrum of the PPLN waveguide used for probe–reference pair generation, the dispersion and length of the fibers (18ps/nm km, 5 km), and detector time uncertainty (83.3 ps) FWHM.

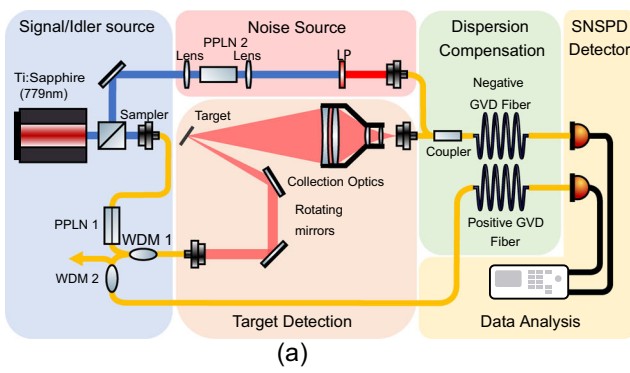

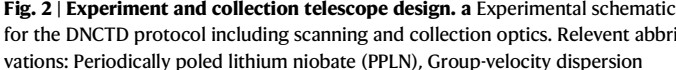

(a)                                                                (b)

**Fig. 2 | Experiment and collection telescope design. a** Experimental schematic for the DNCTD protocol including scanning and collection optics. Relevent abbriviations: Periodically poled lithium niobate (PPLN), Group-velocity dispersion

(GVD), Superconducting nanowire single-photon detector (SNSPD). **b** Purpose-built scanning and collection optics for LiDAR demonstration.

Using the constant velocity of the probe photons in addition to the time delay between probe and reference photons, the depth of a target can be resolved allowing for 3D imaging.

## System design

Photon pairs generated through SPDC exhibit entanglement in the time-frequency degree of freedom. This can manifest itself as a correlation in the detection time of the two photons at separate detectors. This correlation can aid in distinguishing probe light from background noise that has been coupled into the collection optics. Due to the relatively large detector time uncertainty, the full advantage of the temporal correlations cannot be used. It is beneficial then, to measure on a different basis between time and frequency, trading some of the temporal correlation for frequency correlation. In our setup, this is achieved by applying negative dispersion to the probe/noise photon and positive dispersion to the reference photon. The measurable effect of this can be seen in Fig. 1e, f as a slight broadening of the true coincidence peak due to the non-maximal probe-reference entanglement and a substantial broadening of the noise peak. This allows for filtering of the previously indistinguishable uncorrelated noise increasing the SNR. The mathematical details of this improvement are described in Supplementary Note 12. To compare the performance of the new scheme, which we call dispersed non-classical target detection (DNCTD), with previous ones, we also measure the SNR in a classical target detection (CTD) analog, as well as a coincidence-based non-classical target detection (NCTD). The SNRs for these other two schemes are given by

$$\mathrm{SNR_{CTD}} = \frac{\nu \tau_p}{N}, \tag{1}$$

$$\mathrm{SNR_{NCTD}} = \frac{\nu \tau_p \tau_r}{N \nu \tau_r}, \tag{2}$$

where $\nu$ is the pair generation rate, $\tau_p$ and $\tau_r$ are the probe and reference path transmissions, and $N$ is the noise singles rate. The improvement from CTD to NCTD is then given by

$$\mathrm{SNR_{NCTD}} - \mathrm{SNR_{CTD}} = \frac{1}{\nu}. \tag{3}$$

To compare the CTD, NCTD, and DNCTD schemes we must define the SNR as a function of some variable that is independent of additional loss after the noise and probe are combined. To achieve this we define the normalized noise power and normalize probe power to be $\frac{N}{\nu \tau_p}$ and $\frac{\nu \tau_p}{N}$, respectively.

Our scheme differs from QI as demonstrated in[8] in both implementation and assumptions about the environment. To develop an easily implementable LIDAR setup we restrict ourselves to phase-insensitive detection methods for both our scheme and the classical scheme we compare with. We take the environmental noise to be a broadband thermal state however we do not make use of any thermal property of the noise. Instead, our protocol only relies on the noise being spectrally and temporally identical to the probe photon. This assumes any non-identical noise can be filtered through other means. This noise state is independent of the noise state before optimal classical filtering and represents a worst case noise scenario. This constitutes—in principal—a worst case noise scenario because any increase in distinguishablity of the noise from the probe, would reduce the noise power overlapping with the probe photon and be classically filterable. In practice, such optimal filtering is very difficult, especially in the case of broadband SPDC photons. Such filtering will be the subject of future work.

## Experimental Results

The experimental setup used to demonstrate the performance of the DNCTD scheme compared with the NCTD and CTD schemes is shown in Fig. 2a and examined in detail in Supplementary Note 14; however, the relevent parameters are stated here. The probe-reference photons are generated by pumping a Type-0 phase-matched PPLN (LiNbO₃) ridge waveguide. This PPLN waveguide has a length of 10mm. After the pair generation, the pump is filtered using a bulk mixer with <1.2 dB insertion loss at 785 nm and >80 dB isolation at 785 nm. For our experiment a Ti:Sapphire laser was used to pump the PPLN waveguide at 779 nm central frequency. The pump has previously been measured to be 121 ps through an autocorrelation measurement. The probe and reference photons are separated using two coarse wavelength division multiplexors (CWDMs). The first WDM passes the probe photon which is sent to investigate the target. The rejected band is sent to WDM 2 which passes the reference photon.

We characterized the passband of each WDM by sweeping a tunable laser (Santac TSL-550) at 1mW and measuring the transmitted power (Supplementary Fig. 6). For dispersion cancellation, a 5 km long regular SMF-28 fiber was used for the normal dispersion. The dispersion for this fiber around 1550 nm wavelength is 18 ps/(nm km). A standard single-mode dispersion compensation fiber (OFS Low-Loss Wide band LLWBDK: C-168) was used to achieve the anomalous dispersion. The dispersion of this fiber is −172.9 ps/(nm km) and the length was cut to 497.5 to achieve almost an equal magnitude and opposite sign of dispersion. After the dispersive fibers, the probe and reference photons were detected by two superconducting nanowire single-photon detectors (Quantum Opus). For this comparison, we

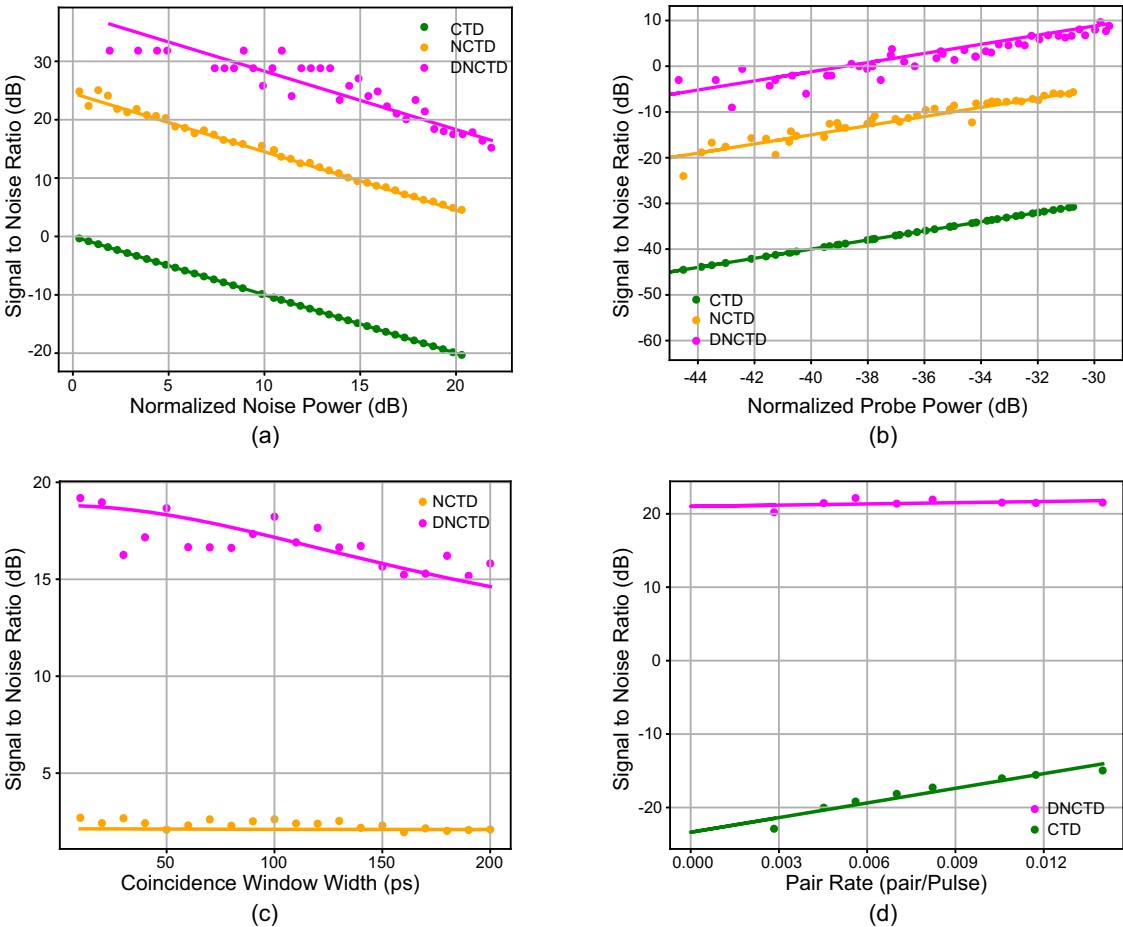

**Fig. 3 | Signal-to-noise ratio comparison of classical target detection (CTD), non-classical target detection (NCTD), and dispersed non-classical target detection (DNCTD) protocols. a** Signal-to-noise ratio for varying noise power. **b** Signal-to-noise ratio for varying probe power. **c** SNR as a function of coincidence window width for DNCTD (magenta) and NCTD (orange). **d** SNR for DNCTD and CTD as a function of SPDC pair rate. The colored lines depict the theoretical predictions for comparison with experimental data.

examine the SNR of each scheme for increasing environmental noise (Fig. 3a), increasing probe channel loss (Fig. 3b), coincidence window width (Fig. 3c), and pump power (Fig. 3d) bypassing the target scanning and collection optics.

First, the normalized noise power was varied while keeping the remaining experimental parameters (pair rate, coincidence window, probe/reference transmission) fixed (Fig. 3a) in a regime convenient for this measurement. The DNCTD SNR is a roughly constant 14.1 dB higher than the NCTD SNR, and the NCTD is a roughly constant 24.5 dB higher than CTD SNR. All three schemes exhibit an approximately linear relationship with the normalized noise power in good agreement with theoretical predictions (colored lines) calculated through Eqs. (1), (2), and the DNCTD theory (Supplementary Note 13).

The SNR was then measured as a function of normalized probe power for the CTD, NCTD, and DNCTD (Fig. 3b) with unique fixed experimental parameters (pair rate, probe/reference transmission, noise power, and coincidence window). The improvement from the CTD to NCTD schemes was calculated from the measured data through Eq. (3) to be an approximately constant 26.0 dB and the improvement from NCTD to DNCTD was measured to be around a constant 12.7 dB yielding a total improvement of 38.7 dB. The SNR exhibits a nearly linear relationship with the normalized probe power for all three schemes as predicted by Eqs. (1) and (2), and the DNCTD theory with the improvements form CTD to NCTD and NCTD to DNCTD remaining approximately constant for all probe powers as predicted.

The SNR was also measured for varying coincidence window widths from 10 to 200 ps (Fig. 3c). For this measurement the noise power was set as high as possible given the variable optical attenuator (VOA) configuration and the pair rate was set large enough to obtain a stable curve at small coincidence windows over a 100 s integration time. Using these parameters, DNCTD can achieve an additional 3.78 dB improvement over the NCTD scheme by changing the coincidence window from 200 to 10 ps. While reducing the coincidence window below the detector uncertainty reduces the number of measured coincidences, this has a more significant impact on the dispersed noise coincidences than the true coincidences resulting in a higher SNR. This can be understood from (Fig. 1f) where the noise distribution is essentially flat in time (for DNCTD) while remaining in a comparatively sharp Gaussian peak for the NCTD scheme. Theoretical calculations performed by integrating the coincidence probabilities (Fig. 1f) over different coincidence window widths are plotted along side the measured data and are in good agreement with the measured results.

Finally, we attempt to show the maximum SNR improvement from CTD to DNCTD possible for our system by varying the pump power, quantified in terms of the SPDC pair generation rate ν (Fig. 3d). The noise power was set as high as possible to achieve the highest SNR difference. The highest measured SNR improvement is 43.1 dB, however much higher improvements are measurable given a higher noise power or longer detection time to adequately estimate the number of noise coincidence counts which become increasingly small at low pair rates. As predicted by Eq. (2), in conjunction with the advantage

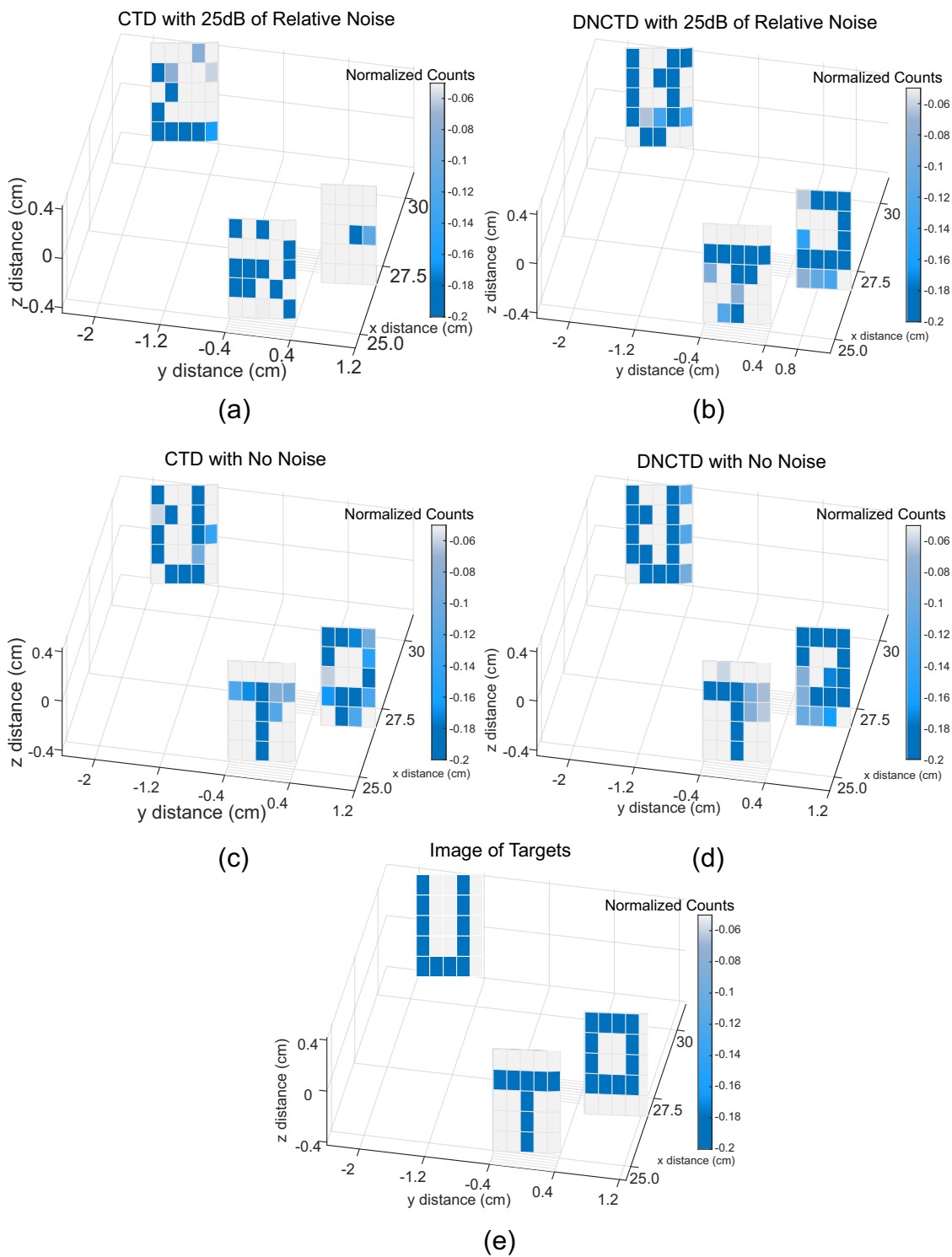

**Fig. 4 | Image scanning result comparison between classical target detection (CTD) and dispersed non-classical target detection (DNCTD) protocols.** **a** CTD scheme with a 25 dB noise background, **b** DNCTD scheme with 25 dB of noise background, **c** CTD with 0 dB noise background, **d** DNCTD scheme with 0 dB noise background, **e** A numerical simulation of ideal imaging of all three letters.

dispersion compensation provides being independent of pair rate, the measured DNCTD SNR is essentially independent of pair rate remaining at $21.45 \pm 1.25$ dB for all measured pair rates. Since the improvement from CTD to NCTD is given by the inverse of the pair rate, the overall improvement from CTD to DNCTD is made largest for minimal pump power, limited only by the system loss and detection efficiency causing the coincidence counts to vanish. Since coincidence counts remain above the noise floor at pair rates where the singles rates are

well below the detector dark counts, we must infer the CTD SNR by a linear fit as a function of pair rate.

Figure 4a–d depicts the scanning results for normalized noise powers 0 and 25dB for both the CTD and DNCTD schemes. Three images (U, O, and T) made of non-reflecting tape placed on a 1inch mirror were scanned at different depths and positions. The intensity of each pixel was measured by using either probe singles counts (CTD) or coincidence counts (NCTD and DNCTD). The targets were angled as to

be not perpendicular to the incident beam to demonstrate the finest depth resolution achievable by our system. In the absence of environmental noise, all three letters are clearly visible in both schemes. However, when the noise is increased well beyond the signal power (25 dB), all three letters become completely indistinguishable in the CTD scheme. In comparison, all three letters remain clearly readable in the DNCTD scheme. In all cases where the letters are visible, the ranging resolution was ±0.09 cm allowing us to not only determine the range of each letter but the difference between the close and far side of each letter.

## Discussion

In conclusion, we proposed, designed, and demonstrated a protocol providing a significant enhancement in target distinguishability from the background noise of a phase-insensitive target-detection scheme. This was achieved by using the non-classical temporal correlations in SPDC photon pairs in conjunction with the relative reduction in noise power in the coincidence window resulting from the non-local dispersion cancellation. We quantified the performance by comparing the SNR of the CTD and DNCTD schemes and found a maximum enhancement of 43.1 dB. We incorporated this scheme with scanning and collection optics to image a target in a noisy environment far beyond what is classically possible at the same probe flux. Our scheme is still limited by a maximum noise flux of around $3.2 \times 10^6$ counts/s or 3.2 MHz above which the detector will saturate. However, due to the pulsed nature of the noise source, this is around 3 times greater than that previously recorded in a temporal correlation-based LiDAR setup in the CW regime[11]. Moreover, it is orders of magnitude larger than the 6.9 kHz noise rate recorded in ref. [19] and 0.4 kHz noise rate recorded in ref. [20]. To explore performance under different noise conditions, We have recently done experiments using a CW noise source and modulation protocol before the application of dispersion. The initial results indicate that our system can maintain an advantage under CW noise conditions as well as determine the range of a target without apriori knowledge of its location. An interesting future study would be to investigate how other non-local effects could be used to further increase the noise-resilience of our target detection scheme.

## Methods

### Varying noise experiment

For the varying noise power experiment, we first collected coincidence data for 61 noise attenuation points each differing by 0.5 dB from 0 to 30 dB as well as one additional point at 60 dB that serves to measure the probe coincidences and singles. Coincidence data were taken for 10 s for each variable optical attenuator (VOA) level (Noise Power Level). For each point, the mean of the coincidence data was taken as the coincidence rate. These data include all the probe and reference singles and coincidence data recorded by the time-tagger during the recording period. While we can estimate the probe singles and coincidences from this attenuation, we also performed the same experiment with the probe light blocked to exclusively record the noise singles and coincidences. The SNR for the CTD case was then calculated for each attenuation with

$$\text{Probe Singles} = \text{Total Singles (60 dB Noise Attenuation)} \quad (4)$$

$$\text{Noise Singles} = \text{Total Singles} - \text{Probe Singles} \quad (5)$$

$$\text{SNR}_{\text{CTD}} = \frac{\text{Probe Singles}}{\text{Noise Singles}} \quad (6)$$

For this set of measurements, the probe singles were obtained from the 60 dB noise attenuation measurement. For the NCTD and

DNCTD schemes the SNR was calculated with

$$\text{True Coincidences} = \text{Coincidence (60 dB Attenuation)} \quad (7)$$

$$\text{Noise Coincidences} = \text{Coincidences (Noise Measurement)} \quad (8)$$

$$\text{SNR}_{\text{NCTD}} = \text{SNR}_{\text{DNCTD}} = \frac{\text{True Coincidences}}{\text{Noise Coincidences}}. \quad (9)$$

The true coincidences were calculated from the 60 dB attenuation measurement and the noise coincidences were measured directly from the measurement where the probe light was blocked.

From the measured data, the pair rate was inferred from equation $\text{SNR}_{\text{NCTD}} = \frac{\nu\tau_p\tau_r}{N_b\nu\tau_r}$ and the singles rates Probe Singles $= \nu\tau_p$ and Ref Singles $= \nu\tau_r$ to be $\nu = 0.0035$ pair/pulse. The probe transmission was found to be 3.3% and the reference transmission 7.2%.

In order to calculate the improvement of the NCTD scheme over the CTD scheme a line was fit to the NCTD data using the variances to weight points. The slope was set to −1 resulting in a vertical offset of 28.788 dB.

### Varying probe experiment

In order to conduct the SNR measurement for varying probe channel loss, the VOA was moved from the noise path to the probe path (before the addition of noise). The loss in the probe arm was varied from 0 to 15 dB in increments of 0.25 dB for a total of 60 measurements with an additional measurement made at 60 dB attenuation to characterize the noise. Each measurement was conducted for 10 s with a coincidence window of 200 ps. This time, the noise light was blocked and another measurement was performed to characterize the probe singles and coincidences. In contrast to the varying noise measurements, it is now the probe singles and coincidences that become increasingly small. This means that, with high attenuation, the variance of the noise is relatively large compared with the probe and so the noise singles and coincidences were calculated by subtracting the 60 dB attenuation measurement from the total. Thus, the SNRs are calculated using

$$\text{SNR}_{\text{CTD}} = \frac{\dfrac{\text{Probe Singles}}{\text{Total Singles} - (\text{Probe Singles} - \text{Background})}}{\dfrac{\text{Probe Singles(60 dB)}}{\text{Total Singles} - (\text{Probe Singles} - \text{Background})}} \quad (10)$$

Further, in order to calculate the probe singles we had to subtract the background counts. This was not an issue in the previous case as the noise singles were calculated by subtracting measured quantities and so the background was implicitly subtracted. The probe singles at 60 dB attenuation were much larger than the background making subtraction insignificant.

$$\text{SNR}_{\text{NCTD}} = \frac{\text{Total Coincidences}-}{\text{Total Coincidences(60 dB)}} - 1 \quad (11)$$

From the measured data, the pair rate was inferred in the same way as the previous section and calculated to be $\nu = 0.00316$ pair/pulse. For the varying probe experiment, the maximum probe transmission was found to be 1.0% and the reference transmission 7.6%. The noise singles were measured to be around a constant 0.0404 counts/pulse.

### Varying coincidence window experiment

Using a coincidence window that fully captures the coincidence peak does not lead to an optimal SNR for the DNCTD scheme. To observe the effect of changing the coincidence window, we performed coincidence measurements with coincidence windows from 10 to 200 ps in

increments of 10 ps. The 200 ps window was chosen as it captures essentially the entire coincidence peak in both the dispersed and non-dispersed regimes. For each coincidence window width, 100 s worth of data were taken and averaged to obtain the singles and coincidence results. The experiment was repeated with the probe disconnected to characterize the noise. The SNR for both the non-dispersed and dispersed experiments were calculated using the equations,

$$SNR_{NCTD} = \frac{\text{Total Coincidences} - \text{Noise Coincidences}}{\text{Noise Coincidences}},$$

$$SNR_{DNCTD} = \frac{\text{Total Coincidences} - \text{Noise Coincidences}}{\text{Noise Coincidences}},$$

For this measurement, the pair rate was set on the order of the background rate and so the pair rate is difficult to estimate accurately as the probe and reference singles counts cannot be well distinguished from background. Nevertheless, the coincidence counts are still easily countable as the coincidence background is essentially zero and this measurement does not compare with the CTD scheme and the knowledge of the singles does not change the result. For this experiment, the noise singles were 0.019 counts/pulse.

**Varying pump power experiment**

In order to achieve the largest possible SNRs difference between the CTD and DNCTD schemes, we reduce the SPDC pair generation rate as low as possible while using a 10 ps coincidence window. The SNR was measured for eight different pump power levels (adjusted with a continuously variable attenuation wheel). Each measurement consisted of 500 s worth of data for a 10 ps coincidence window with no noise attenuation, 250 s worth of data for a 10 ps coincidence window with 60 dB of noise attenuation, 250 s worth of data with a 200 ps coincidence window wtih 60 dB of noise attenuation. The first two measurements were used to characterize the total, probe, and noise coincidences and singles data while the last measurement is used to infer the pair rate. The SNR was calculated using the equations

$$\text{Probe Singles} = \text{Singles (60 dB Noise Attenuation)} - \text{Background Singles} \quad (12)$$

$$\text{Noise Singles} = \text{Singles (0 dB Noise Attenuation)} - \text{Probe Singles} \quad (13)$$

$$\text{True Coincidences} = \text{Coincidences (60 dB Noise Attenuation)} \quad (14)$$

$$\text{Noise Coincidences} = \text{Coincidences (0 dB Noise Attenuation)} \quad (15)$$

$$SNR_{CTD} = \frac{\text{Probe Singles}}{\text{Noise Singles}} \quad (16)$$

$$SNR_{NCTD/DNCTD} = \frac{\text{True Coincidences}}{\text{Noise Coincidences}} \quad (17)$$

To compare each of the data points a metric for the pump power was needed. The pair rate suffices well however to calculate it the full number of coincidences within the peak must be known. That is, the coincidence counts using the 10 ps window cannot be used. Using the 200 ps coincidence window measurement, the pair rate is calculated according to the equations

$$\text{Reference Singles} = \text{Reference Singles} - \text{Background Reference Singles} \quad (18)$$

$$\text{Probe Singles} = \text{Probe Singles} - \text{Background Probe Singles} \quad (19)$$

$$\text{True Coincidences} = \text{Coincidences}(60 \text{ dB Noise Attenuation}, \\ 200 \text{ ps Coincidence Window}) \quad (20)$$

$$\text{Pair Rate} = \frac{\text{Probe Singles} \times \text{Reference Singles}}{\text{Coincidences}} \quad (21)$$

## Data availability
The data generated during this study are available from the corresponding author upon request.

## Code availability
The code used for the purposes of analyzing data is also available from the corresponding author upon request.

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

## Acknowledgements

This work was supported by the Innovation for Defence Excellence and Security (IDEaS) program and the Natural Sciences and Engineering Research Council of Canada (AH).

## Author contributions

P.B. performed sections of the theoretical analyses, conducted the experiment, and wrote the manuscript. H.L. developed the theoretical underpinning of the protocol and aided in performing the experiments. G.P. designed and built the scanning collection optics and aided in performing the experiments. Y.Z., Z.L., and M.I. aided in performing the experiments. A.M. supervised the project and participated in writing and vetting the manuscript.

## Competing interests

The authors declare no competing interests.
