## [Peer Review File · Nature Communications]

Quantum and Non-local Effects Over over 40dB Noise Resilience Advantage Towards Quantum LiDARREVIEWER COMMENTS

Reviewer #1 (Remarks to the Author):

In the recent years a great advance in experimental quantum optics has made it possible to develop many quantum sensing techniques allowing to achieve a strong improvement in sensitivity compared to classical protocols. Unfortunately, many of these quantum enhanced protocols are very sensible to noise and losses. Quantum illumination represents an exception on this scenario being a very robust quantum protocol for object detection.

Based on the quantum illumination protocol, the authors propose and experimentally demonstrate a quantum enhanced LIDAR promising to provide a significant improvement in the target distinguishability from the background noise in practical scenarios. The novelty element of this scheme is represented by the exploitation of a scheme of dispersion cancellation based on the quantum correlation between the probe and the reference.

In this work the authors inject a noise into the probe beam having the same spectral and temporal distribution as the probe. Then, two optical fibres, with positive and negative GVD respectively, are used to introduce a dispersion in both the probe and the reference signal. Due to the quantum temporal correlations between the probe and the reference beam, temporal coincidences undergo to a dispersion cancellation, otherwise, events that are just classically correlated, such as accidental coincidences, undergo a broadening of the coincidence peak, allowing to separate the signal from the noise in post-selection.

In principle the technique looks valid and the data obtained in the experiment are convincing. The paper is quite clear and well written and the additional material is exhaustive.

However, the strength of the work is represented precisely by the possibility of applying this scheme in a practical scenario. Because, in principle, this technique does not introduce any improvement from a fundamental point of view, but it is only aimed to overcoming the limited temporal resolution of the detectors.

The technique works because the authors considered noise that has the same temporal distribution as the signal. With a continuous or temporally chaotic noise, which is more similar to practical scenarios, the spurious coincidences are already temporally scattered and therefore the dispersion introduced by the fibres has no useful effect.

Of course, the authors state that any temporal mode different from the signal mode can in principle be filtered with classical techniques before the dispersion. This is true; it is possible to use some ultrafast optical element to select only the temporal modes corresponding to the signal. However, it is necessary to know a priori the arrival time of the signal to classically isolate the right temporal window. This is possible only if you know a priori the position of the target vanishing any practical application as a LIDAR.

Therefore, if my doubt is correct, the impact of this work is limited by the fact that the technique is only useful in well-controlled laboratory scenarios and not for practical applications as claimed in the paper.

Reviewer #2 (Remarks to the Author):

The authors demonstrate their approach to reducing noise from anonalous signals in LiDAR, specifically relying on frequency anti-correlation from entangled photon pair in a quantum LiDAR scheme. Overall I find this to be an interesting and innovative idea with a very clearly demonstrated advantage. There are some areas, however that I think should be cleared up before the paper is

published.

- Throughout the manuscript, the authors refer to additive, unwanted radiation as "noise". This word by itself can take many different meanings, especially with the numerous reference to phase-sensitive LiDAR. The authors should clarify what this means in the context of their work, I think this only needs to be done once at the beginning.

- Presumably the dispersion in probe and reference arms needs to be equal but opposite. If so, it should be specified. As far as I can see there is no reference to this in the text.

- Much of the text in fig 1 is too small to read.

- For the caption in Fig.1 relating to the simulations they discuss photon pair generation in a waveguide but for the experiment a bulk crystal is used. What is the reason for choosing to use waveguides here?

- There are some key details that, in my opinion, should be made clear in the main body of the text. For example, how long is the crystal used to generate the photons? What's the material? Is there any spectral filtering? What is the dispersion of the fibers (better yet if they can provide exact details of the fibers)? What's the laser pulse duration?

Some of this can be inferred from the details in fig.1 and some is in the supplementary materials however I feel it is worth stating these key points in the main text clearly as they are critical for understanding the degree of entanglement, the temporal binning that's needed etc.

- Just before the Experimental Results section, the authors say "we do not take the background noise to be thermal radiation but assume that we can optimally filter the coupled noise, removing anything that is not spectrally identical to the probe photon." I find this statement confusing, "thermal" suggests something about the photon arrival statistics (preferentially bunched photons) and not the spectral properties i.e. one could have a light source with the same spectral properties as the probe but could still be bunched. Could the authors clarify what they mean?

- In Fig.3, the uncertainty in the proposed SNR for the proposed DNCTD scheme seems much higher than the others. What is the reason for this? Is it because it's a much lossier process?

- The acronym VOA is not defined

- The authors discuss saturation levels, but this is surely detector specific. This should be stated.

- One of the references is missing in the supplementary, section 1.6

- When using photon pairs, the relative arrival time can be used for the timing aspect of LiDAR instead of a pulsed laser (see e.g. Liu et al. Optica Vol. 6, Issue 10, pp. 1349-1355 (2019)). The spectral entanglement for a CW pump laser would also arguably be stronger as this influences the width of the spectral joint probability distribution. Is there any reason the authors opted for a femtosecond laser instead?

Reviewer #3 (Remarks to the Author):

This manuscript shows some ostensibly very impressive results showing the possible advantages obtained in an imaging context using quantum illumination. It exploits a suite of well-known quantum optical effects based on the enhanced quantum correlations of twin beams to produce images of objects significantly better than they can do classically. The manuscript is clearly written and the supplementary material provides a lot of detail on the methods used.

The basis of coincidence click quantum illumination schemes is essentially mode-matching of the signal and idler clicks. If you can exclude from signal a large fraction of the noise by detecting in coincidence with the idler you win. Normally this is done only with temporal coincidences and frequency modes. The quantum correlations found in correlated pairs produced from say SPDC allow this easily. The main extra thing in this manuscript is that they cancel some of the dispersion introduced in the probe signal arm by using an opposite dispersion in the idler arm – a counterintuitive effect known as nonlocal dispersion cancellation, predicted by Franson. This allows better mode overlap between the signal and idler arms and increases the fraction of nonaccidental coincidences. Hence their significant improvement in imaging.

I have some points to consider.

1. Firstly based on the title of the manuscript. The word LiDAR is used. To me this conjures up object detection and ranging. The manuscript seems more about imaging and not about object detection – certainly not ranging. Imaging and object detection are easy to relate, but ranging is a free space out-of-lab problem where the effective reflectivity can easily be lower than one part per billion. This is the most important quantum LiDAR problem and is not one that can easily be addressed by lab-based experiments. The authors do not seem to provide an overall detection probability (effective reflectivity) for a photon sent in to interrogate the target, which should be something like target intrinsic target reflectivity \times collection efficiency \times coupling losses \times detector q.e. The authors should comment on the above, and/or provide a use-case for their (still impressive) results. If that use case is not actually some form of LiDAR they should consider changing to “quantum illumination” in the title.

2. The same goes for the dispersion cancellation that is part of the main thrust of the manuscript. Could this reasonably help object identification outside the lab, in anything other than, say, a microscopy experiment?

On a much more minor note, there are also dispersion cancellation effects based on frequency-dependence of the loss that can change coincidence pulse widths nonlocally. Would this help their scheme or is the overall probe-arm loss frequency-independent?

3. The other main point is on limits. The claim is ~ 40 dB improvement over a phase insensitive classical target detection scheme. I do not doubt it, but this will not be where the classical limit is. The best possible discrimination between the two possible detected states (target there or not) will be given by the Helstrom probability bound between them. The authors have shown that they are in a regime where they can easily beat *their* classical illumination, but not whether their improvement is better than classically allowed. It is likely that at the lowest probe powers in say fig 3b they beat the classical limit, but will be less clear for higher powers, because click detection is nonideal for the task of state discrimination. This point should be dealt with mostly in the supplementary material, but at least mentioned in the main text.

I believe the work will be significant for the field, and if the authors can address the above comments I support publication.

Reviewer 1 Replies

Comment 1.)

In the recent years a great advance in experimental quantum optics has made it possible to develop many quantum sensing techniques allowing to achieve a strong improvement in sensitivity compared to classical protocols. Unfortunately, many of these quantum enhanced protocols are very sensible to noise and losses. Quantum illumination represents an exception on this scenario being a very robust quantum protocol for object detection.

Based on the quantum illumination protocol, the authors propose and experimentally demonstrate a quantum enhanced LIDAR promising to provide a significant improvement in the target distinguishability from the background noise in practical scenarios. The novelty element of this scheme is represented by the exploitation of a scheme of dispersion cancellation based on the quantum correlation between the probe and the reference.

In this work the authors inject a noise into the probe beam having the same spectral and temporal distribution as the probe. Then, two optical fibres, with positive and negative GVD respectively, are used to introduce a dispersion in both the probe and the reference signal. Due to the quantum temporal correlations between the probe and the reference beam, temporal coincidences undergo to a dispersion cancellation, otherwise, events that are just classically correlated, such as accidental coincidences, undergo a broadening of the coincidence peak, allowing to separate the signal from the noise in post-selection.

In principle the technique looks valid and the data obtained in the experiment are convincing. The paper is quite clear and well written and the additional material is exhaustive.

However, the strength of the work is represented precisely by the possibility of applying this scheme in a practical scenario. Because, in principle, this technique does not introduce any improvement from a fundamental point of view, but it is only aimed to overcoming the limited temporal resolution of the detectors.

The technique works because the authors considered noise that has the same temporal distribution as the signal. With a continuous or temporally chaotic noise, which is more similar to practical scenarios, the spurious coincidences are already temporally scattered and therefore the dispersion introduced by the fibres has no useful effect.

Of course, the authors state that any temporal mode different from the signal mode can in principle be filtered with classical techniques before the dispersion. This is true; it is possible to use some ultrafast optical element to select only the temporal modes corresponding to the signal. However, it is necessary to know a priori the arrival time of the signal to classically isolate the right temporal window. This is possible only if you know a priori the position of the target vanishing any practical application as a LIDAR.

Therefore, if my doubt is correct, the impact of this work is limited by the fact that the technique is only useful in well-controlled laboratory scenarios and not for practical applications as claimed in the paper.

Response:

This comment raises a very important point. To address the concerns of the reviewer, within the time constraints and laboratory constraints we performed a proof-of-principal experiment to demonstrate that dispersion cancellation can provide increased noise resilience in practical applications as well as well-controlled laboratory scenarios. To do this we first describe a protocol for ranging a target without apriory knowledge of its range. We then perform a proof-of-concept experiment demonstrating the potential of this technique. We note upfront that the results obtained are not intended to represent the best such a protocol can achieve. Rather, even with the time and laboratory equipment limitations imposed by replying in a timely manner, we can obtain an advantage using our dispersion cancellation protocol with the additional conditions mentioned by the reviewer. We have adjusted the text in the manuscript to reflect the difficulty involved in dealing with noise as described by the reviewer, as well as the promising results of our proof-of-principal experiment.

While CW noise is not the only type of environmental noise, to address the point raised by the reviewer, we shall take the noise to be CW or chaotic. Other forms of noise may arise due to, for example, adverse third parties attempting to jam detection. The coincidences between such a CW background and the reference photons would indeed be temporally scattered and dispersion would not provide an advantage as the noise power of such a noise signal is increased, the certainty with which the coincidence peak corresponding to the returned signal can be located decreases. In this situation, some ultrafast optical element should be used to pick out the temporal mode corresponding to our signal so that dispersion may be used to reduce the noise power overlapping with the signal and increase the certainty with which the peak can be located. In the situation where the probe power is sufficiently low and the noise sufficiently high, that a coincidence peak cannot be located (illustration in Fig 1a and Fig 1b), one cannot without apriori knowledge of the target location, know how to modulate the incoming signal correctly and must perform a search for the peak. We know describe a protocol for such a search

Protocol Description

We assume that the light sent to interrogate the target is pulsed with repetition rate τ_{rep} . This requirement can in fact be relaxed and a CW probe can be used in principle. To locate this peak initially it is not necessary to modulate the incoming signal at the photon signal duration ($\sim 20\text{ps}$ in our experiment based on the phase-matching bandwidth). It is only necessary to reduce the noise floor enough to effectively locate this peak. This process is illustrated in Fig 1 and is as follows:

- 1.) Choose a duty cycle D (for example $\frac{1}{4}$ in Fig 1) and modulate the signal between pulses with a square pulse with this duty cycle.

- 2.) Break up the period of repetition into subsections with duration $D \times \text{period}$.

3.) Allow the non-local cancellation of dispersion to broaden the noise within the repetition rate.

4.) Shift the phase of the pulse so that the unattenuated portion of the probe occupies the next section of the period (Fig1d).

5.) Repeat until the peak is visible.

6.) If the peak is not visible, the duty cycle can be further reduced.

7.) Once the coincidence peak is located, the coincidence bin number can be used, in combination with a calibration coincidence measurement at a known distance to measure the range of the target.

Note also, once the peak is located, the duty cycle may be further reduced and one can ‘lock on’ to the coincidence peak.

Fig 1: Illustration of modulation scheme for peak location without apriori knowledge of target location. (a) Simulated raw coincidence data where peak is not visible. (b) Modulation of the first possible bin before dispersion. (c)

Modulation of the first bin after dispersion of noise to be equal over the period. (c) Same as (a). (d) Modulation of the second bin before dispersion. (e) Modulation of the second bin after dispersion.

Experimental Description

To further probe the validity of this technique for peak location, we now document a proof-of-principal experiment implementing this search technique. The experimental setup is very similar to that of the manuscript and is depicted in Fig 2. The same Ti:Sapphire laser was used to pump the same PPLN ridge-waveguide, and the same WDMs were used to separate the probe and reference photons. The reference photon was directed directly towards a detector, with the inclusion of dispersion for one set of measurements. The probe is combined with CW noise originating from an LED whose spectrum was filtered using a Alnair Labs BVF-200CL Bandwidth-Variable Tunable Filter to match that of the probe photon as measured by the WDM (Supplementary Figure 6: $\sim 1560\text{-}1580\text{nm}$). The combined probe-noise path was directed towards a Lithium Niobate electro optic modulator. The modulator was controlled by a Highland Technology T-130 Electrical/ Optical Modulation Driver triggered by the electrical trigger signal from the Ti: Sapphire laser. The modulated probe-noise was then directed through, optionally, a dispersion shifted fiber towards the superconducting nanowire single-photon detector (SNSPD).

Due to current limitations on the available dispersion, modulation duty cycle, and loss from the modulation system as well as to reply in a timely fashion, we opted not to measure the range of a free space target. Instead, we prove that dispersion can be used to locate a coincidence peak in an environment plagued by CW noise strong enough to completely obscure the peak without the non-local dispersion cancellation technique. With more time, and better collection optics, this scheme could easily be extended to range a free-space target.

Fig 2: Experimental schematic for target ranging in a noisy environment.

Experimental Results

Due to the limited dispersion and modulation pulse width, we were only able to disperse the noise-reference coincidences by 2.26 times for a 1ns modulated noise pulse width. Thus, to observe improved peak distinguishability---and subsequent ranging---we used the lowest pulse width achievable (with reasonable jitter) from our modulator driver at 1ns allowing for the largest possible fractional peak broadening. The modulator delay was then swept from 0 to 12ns. For each 1ns delay increment, coincidence data was recorded for 30s. With more time and better equipment (such as the Berkely Nucleonics 765 800 MHz Fast Rise Time Pulse Generator which can achieve generate pulses with duration 300ps with ~10ps jitter), this recording time could be substantially reduced allowing for faster searching times.

We recorded histogram data from -40 to 40ns corresponding to 23.98m (Figure 3 (a) and (b)). For data processing we only examine the 6 coincidence peaks that remain within the window for all delays corresponding to 22.48m. We note however, that the true limitation here is imposed by the time-tagging device and is well beyond 80ns. The coincidence histogram with of 80ns was only for the purpose of visualizing the results and proof-of-principal.

Fig 4: (a) Coincidence histogram data for zero delay and 1ns modulated pulse width. (b) Coincidence histogram data with dispersion cancellation for zero delay and 1ns modulated pulse width.

The noise level was adjusted so that, in the absence of dispersion, the true coincidence peak was no longer distinguishable from noise. Precisely, using a 1ns window the classical SNR as defined in the main manuscript is $-30.73dB$ and the quantum non-dispersed SNR $-17.5dB$. The indistinguishability is shown in figure 4 (a) where the true coincidence peak takes on a value 1.77 time standard deviations above the mean.

For each delay, the coincidence peak height was calculated for each peak in the histogram and normalized by the mean peak height (Figure 4 (a) and (b)). Under the assumption that the noise is CW, a peak significantly above 1 indicates the location of the true coincidence peak. At a delay of 11.5ns pulse 3 takes on a value 3.58 standard deviations (one standard deviation 0.0116) above the mean (1). No such point exists in the non-dispersed data with the true coincidence peak (pulse 3 at 11.5ns delay) taking a value of 1.14. The standard deviation for the non-dispersed data is 0.0079.

Fig 4: (a) Normalized coincidence peak data for modulation delays 0-12ns and a modulation width of 1ns. (b) Normalized dispersed coincidence peak data for modulation delays 0-12ns and a modulation width of 1ns.

Using limited dispersion and limited modulation pulse width, we were able to recover a statistically significant coincidence peak from an environment plagued by noise where, without modulation and dispersion, the coincidence peak was indistinguishable. We believe this proof-of-principal experiment confirms that it is possible to detect and range a target's location without a priori knowledge of its range when the environmental noise is continuous. While these results are meant only to demonstrate the principle of this search technique, a more robust experimental setup could easily be developed given more time. Such a setup could significantly enhance the contrast of the true peak from the noise peaks after dispersion. The following are important considerations for such an apparatus:

- The amount of dispersion experienced by the probe/noise and reference paths must be increased without significant increase in loss. One of the main limitations that prevented us from adding more fiber to our setup was the mode mismatch between the anomalous dispersion fiber and the rest of the setup. Adding a second of such fibers would significantly increase the loss, prohibiting the demonstration of the protocol. A less lossy way of adding more dispersion to the setup would allow the system a much larger noise tolerance.
- To implement scanning and collection optics, the overall system loss must be reduced. This would involve redesign of the telescope, upgrading the various optical elements in the probe path, and the construction of a more permanent apparatus.
- The modulator driver's output pulse experienced a high jitter when the modulation pulse width was below 1ns. Smaller duty cycles will lead to a more significant dispersion and higher contrast between the true peak and noise peaks. A good piece of equipment for such purposes is the Berkely Nucleonics 765 800 MHz Fast Rise Time Pulse Generator which can generate pulses with duration 300ps with ~10ps jitter.
- The SPDC source should also be tailored to the specific needs of the problem. The efficiency of our waveguide was not fully utilized as we were pumping a long waveguide with a femtosecond

laser. A more optimal pair could lead to both higher flux, and a higher degree of time-frequency entanglement allowing for more optimal dispersion cancellation.

Reviewer 2 Replies

Comment 1.)

“Throughout the manuscript, the authors refer to additive, unwanted radiation as "noise". This word by itself can take many different meanings, especially with the numerous reference to phase-sensitive LiDAR. The authors should clarify what this means in the context of their work, I think this only needs to be done once at the beginning.”

Response:

Thank you very much for your suggestion. This is a valuable point, and we should be more clear as to what noise means for our context. We suggest the definition,

“By noise we refer to the portion of the returned light that is uncorrelated with the presence or absence of the target. This can include other illuminating sources, natural light, jamming signals, etc.”

We have made this change in our revised manuscript.

Comment 2.)

“Presumably the dispersion in probe and reference arms needs to be equal but opposite. If so, it should be specified. As far as I can see there is no reference to this in the text.”

Response:

Thank you for this good catch. Indeed, it is correct that the dispersion in the probe and reference arms needs to be equal but opposite in sign. We have made the relevant changes in our manuscript.

Comment 3.)

“Much of the text in fig 1 is too small to read.”

Response:

We have adjusted the text size to make it more readable in the revised manuscript.

Comment 4.)

“For the caption in Fig.1 relating to the simulations they discuss photon pair generation in a waveguide but for the experiment a bulk crystal is used. What is the reason for choosing to use waveguides here?”

Response:

We apologize for the confusing details in the figure. To clarify: we used a PPLN waveguide to generate photon pairs and a bulk PPLN crystal to generate the noise. We have made this clearer in the figure now.

The use of a PPLN waveguide was predominantly dictated by what was available to us at the time of experiment. Nevertheless, a PPLN waveguide is suitable for our application in terms of high source brightness and low output coupling loss. This helps achieving higher SNRs for both the classical and non-classical measurements and making our statistical analysis more accurate.

Comment 5.)

“There are some key details that, in my opinion, should be made clear in the main body of the text. For example, how long is the crystal used to generate the photons? What's the material? Is there any spectral filtering? What is the dispersion of the fibers (better yet if they can provide exact details of the fibers)? What's the laser pulse duration? Some of this can be inferred from the details in fig.1 and some is in the supplementary materials however I feel it is worth stating these key points in the main text clearly as they are critical for understanding the degree of entanglement, the temporal binning that's needed etc.”

Response:

Thank you for your suggestion. We now clearly list the experiment details in section IV of the main manuscript. In this section we mention some measurements to characterize the WDMs used to split the probe-reference photons and an autocorrelation measurement used to characterize the pump pulse. These results were put in section 4.1 of the supplementary material.

Comment 6.)

Just before the Experimental Results section, the authors say "we do not take the background noise to be thermal radiation but assume that we can optimally filter the coupled noise, removing anything that is not spectrally identical to the probe photon." I find this statement confusing, "thermal" suggests something about the photon arrival statistics (preferentially bunched photons) and not the spectral properties i.e. one could have a light source with the same spectral properties as the probe but could still be bunched. Could the authors clarify what they mean?

Response:

Sorry for the misconception. Indeed, our noise source is a broadband thermal state since it is generated from the SPDC process in the bulk PPLN (but with broken entanglement due to attenuation). However, we did not make any use of the thermal property of the noise light because the protocol only relies on correlations in the time-frequency degree of freedom. We have made this point more clear now in section III of the main manuscript

Comment 7.)

In Fig.3, the uncertainty in the proposed SNR for the proposed DNCTD scheme seems much higher than the others. What is the reason for this? Is it because it's a much lossier process?

Response:

Thank you for noticing this. The short explanation to this artifact is that when the noise power is low, the calculation of the SNR becomes very inaccurate.

The SNR for the DNCTD scheme is calculated as the ratio of probe-reference coincidence to noise-reference coincidences as a function of either the normalized noise power or normalized probe power.

For the first case of varying noise power, due to the reduction in noise coincidences in the DNCTD scheme the relative variance of these noise coincidences correspondingly increases. This in turn substantially increases the variance of the SNR at very high SNRs. This can be seen from figure 3A (plotted below for convenience) where the variance of the DNCTD increases from right to left. Similar effect can also be observed for the NCTD scheme when the noise power is low. The reason why the DNCTD curve has larger fluctuation than the NCTD curve is that in the DNCTD scheme, the noise coincidence rate is lower due to correlation enhancement. This results in the SNR being completely infinite for some noise power data points for example from 5 to 7dB.

(a)

When the SNR is measured as a function of probe power (probe loss), the variance increases for very low probe powers. This is analogous to the previous case however instead of the variance of the noise coincidences increasing the variance of the probe coincidences increases due to a low number of counts.

While we can always reduce the variance by taking a longer integration time, this would only serve to extend our x-axis as no matter the experimental integration time we can use a noise power or probe power that has significant variance. Therefore, we opted to include the high variance points in our data to depict the effect of exceedingly low noise power, and exceedingly low probe power.

(b)

Comment 8.)

The acronym VOA is not defined

Response:

Thank you very much for this catch. The definition variable optical attenuator has been added to the revised manuscript

Comment 9.)

The authors discuss saturation levels, but this is surely detector specific. This should be stated.

Response:

Thank you for mentioning this point. The saturation level is indeed detector specific and predominantly depends on the detector dead time. We have updated our statement in the abstract to reflect this. In the condition that the detector is saturating, modulation followed by dispersion may be used to reduce the noise power within the detector dead time and stop detector saturation. To achieve this, the modulation rate must be longer than the detector dead time so that we may disperse the incident noise outside the detector deadtime window. For our experiment, given a modulated pulse duration of 20ps (on the order of the probe photon duration), this would then lead to a saturation resilience of $\frac{25ns}{20ps} = 1250$.

Comment 10.)

One of the references is missing in the supplementary, section 1.6

Response:

Thank you very much for this good catch. We have added the proper citation in the supplementary material.

Comment 11.)

When using photon pairs, the relative arrival time can be used for the timing aspect of LiDAR instead of a pulsed laser (see e.g. Liu et al. Optica Vol. 6, Issue 10, pp. 1349-1355 (2019)). The spectral entanglement for a CW pump laser would also arguably be stronger as this influences the width of the spectral joint probability distribution. Is there any reason the authors opted for a femtosecond laser instead?

Response:

This is a good point. We do use the relative arrival time of the probe and reference photons for depth resolution in our setup and the spectral entanglement for a CW pump laser would indeed be stronger. To implement dispersion cancellation in the CW regime one would have to chop the incoming signal and noise prior to the dispersion to allow the noise room to disperse. To observe the optimal results a fast modulator $\sim 50\text{-}100\text{GHz}$ would be required which is a piece of equipment that we currently do not have. Moreover, this has the effect of increasing loss in the probe arm while having the experiment resemble that where a pulsed pump is used. We use a femtosecond pump to achieve the same effect as modulation of the received signal without the need for such modulators.

However, to reply to a question from another reviewer, we performed a proof of principal experiment where a modulator was used to chop CW noise before the application of dispersion cancellation. While we did not use a CW pump, such a pump could be used in this protocol. The description and results of this experiment are described below.

Protocol Description

We assume that the light sent to interrogate the target is pulsed with repetition rate τ_{rep} . This requirement can in fact be relaxed and a CW probe can be used in principle. To locate this peak initially it is not necessary to modulate the incoming signal at the photon signal duration ($\sim 20\text{ps}$ in our experiment based on the phase-matching bandwidth). It is only necessary to reduce the noise floor enough to effectively locate this peak. This process is illustrated in Fig 1 and is as follows:

- 1) Choose a duty cycle D (for example $\frac{1}{4}$ in Fig 1) and modulate the signal between pulses with a square pulse with this duty cycle.

- 2) Break up the period of repetition into subsections with duration $D \times \text{period}$.

- 3) Allow the non-local cancellation of dispersion to broaden the noise within the repetition rate.

- ↓
- 4) Shift the phase of the pulse so that the unattenuated portion of the probe occupies the next section of the period (Fig1d).

↓

 - 5) Repeat until the peak is visible.

↓

 - 6) If the peak is not visible, the duty cycle can be further reduced.

↓

 - 7) Once the coincidence peak is located, the coincidence bin number can be used, in combination with a calibration coincidence measurement at a known distance to measure the range of the target.

Note also, once the peak is located, the duty cycle may be further reduced and one can ‘lock on’ to the coincidence peak.

Fig 1: Illustration of modulation scheme for peak location without apriori knowledge of target location. (a) Simulated raw coincidence data where peak is not visible. (b) Modulation of the first possible bin before dispersion. (c) Modulation of the first bin after dispersion of noise to be equal over the period. (d) Same as (a). (e) Modulation of the second bin before dispersion. (f) Modulation of the second bin after dispersion.

Experimental Description

To further probe the validity of this technique for peak location, we now document a proof-of-principal experiment implementing this search technique. The experimental setup is very similar to that of the manuscript and is depicted in Fig 2. The same Ti:Sapphire laser was used to pump the same PPLN ridge-waveguide, and the same WDMs were used to separate the probe and reference photons. The reference photon was directed directly towards a detector, with the inclusion of dispersion for one set of measurements. The probe is combined with CW noise originating from an LED whose spectrum was filtered using a Alnair Labs BVF-200CL Bandwidth-Variable Tunable Filter to match that of the probe photon as measured by the WDM (Supplementary Figure 6: $\sim 1560\text{-}1580\text{nm}$). The combined probe-noise path was directed towards a Lithium Niobate electro optic modulator. The modulator was controlled by a Highland Technology T-130 Electrical/ Optical Modulation Driver triggered by the electrical trigger signal from the Ti: Sapphire laser. The modulated probe-noise was then directed through, optionally, a dispersion shifted fiber towards the superconducting nanowire single-photon detector (SNSPD).

Due to current limitations on the available dispersion, modulation duty cycle, and loss from the modulation system as well as to reply in a timely fashion, we opted not to measure the range of a free space target. Instead, we prove that dispersion can be used to locate a coincidence peak in an environment plagued by CW noise strong enough to completely obscure the peak without the non-local dispersion cancellation technique. With more time, and better collection optics, this scheme could easily be extended to range a free-space target.

Fig 2: Experimental schematic for target ranging in a noisy environment.

Experimental Results

Due to the limited dispersion and modulation pulse width, we were only able to disperse the noise-reference coincidences by 2.26 times for a 1ns modulated noise pulse width. Thus, to observe improved

peak distinguishability---and subsequent ranging---we used the lowest pulse width achievable (with reasonable jitter) from our modulator driver at 1ns allowing for the largest possible fractional peak broadening. The modulator delay was then swept from 0 to 12ns. For each 1ns delay increment, coincidence data was recorded for 30s. With more time and better equipment (such as the Berkely Nucleonics 765 800 MHz Fast Rise Time Pulse Generator which can achieve generate pulses with duration 300ps with ~10ps jitter), this recording time could be substantially reduced allowing for faster searching times.

We recorded histogram data from -40 to 40ns corresponding to 23.98m (Figure 3 (a) and (b)). For data processing we only examine the 6 coincidence peaks that remain within the window for all delays corresponding to 22.48m. We note however, that the true limitation here is imposed by the time-tagging device and is well beyond 80ns. The coincidence histogram with of 80ns was only for the purpose of visualizing the results and proof-of-principal.

Fig 4: (a) Coincidence histogram data for zero delay and 1ns modulated pulse width. (b) Coincidence histogram data with dispersion cancellation for zero delay and 1ns modulated pulse width.

The noise level was adjusted so that, in the absence of dispersion, the true coincidence peak was no longer distinguishable from noise. Precisely, using a 1ns window the classical SNR as defined in the main manuscript is $-30.73dB$ and the quantum non-dispersed SNR $-17.5dB$. The indistinguishability is shown in figure 4 (a) where the true coincidence peak takes on a value 1.77 time standard deviations above the mean.

For each delay, the coincidence peak height was calculated for each peak in the histogram and normalized by the mean peak height (Figure 4 (a) and (b)). Under the assumption that the noise is CW, a peak significantly above 1 indicates the location of the true coincidence peak. At a delay of 11.5ns pulse 3 takes on a value 3.58 standard deviations (one standard deviation 0.0116) above the mean (1). No such point exists in the non-dispersed data with the true coincidence peak (pulse 3 at 11.5ns delay) taking a value of 1.14. The standard deviation for the non-dispersed data is 0.0079.

Fig 4: (a) Normalized coincidence peak data for modulation delays 0-12ns and a modulation width of 1ns. (b) Normalized dispersed coincidence peak data for modulation delays 0-12ns and a modulation width of 1ns.

Using limited dispersion and limited modulation pulse width, we were able to recover a statistically significant coincidence peak from an environment plagued by noise where, without modulation and dispersion, the coincidence peak was indistinguishable. We believe this proof-of-principal experiment confirms that it is possible to detect and range a target's location without a priori knowledge of its range when the environmental noise is continuous. While these results are meant only to demonstrate the principle of this search technique, a more robust experimental setup could easily be developed given more time. Such a setup could significantly enhance the contrast of the true peak from the noise peaks after dispersion. The following are important considerations for such an apparatus:

- The amount of dispersion experienced by the probe/noise and reference paths must be increased without significant increase in loss. One of the main limitations that prevented us from adding more fiber to our setup was the mode mismatch between the anomalous dispersion fiber and the rest of the setup. Adding a second of such fibers would significantly increase the loss, prohibiting the demonstration of the protocol. A less lossy way of adding more dispersion to the setup would allow the system a much larger noise tolerance.
- To implement scanning and collection optics, the overall system loss must be reduced. This would involve redesign of the telescope, upgrading the various optical elements in the probe path, and the construction of a more permanent apparatus.
- The modulator driver's output pulse experienced a high jitter when the modulation pulse width was below 1ns. Smaller duty cycles will lead to a more significant dispersion and higher contrast between the true peak and noise peaks. A good piece of equipment for such purposes is the Berkely Nucleonics 765 800 MHz Fast Rise Time Pulse Generator which can generate pulses with duration 300ps with ~10ps jitter.
- The SPDC source should also be tailored to the specific needs of the problem. The efficiency of our waveguide was not fully utilized as we were pumping a long waveguide with a femtosecond

laser. A more optimal pair could lead to both higher flux, and a higher degree of time-frequency entanglement allowing for more optimal dispersion cancellation.

Reviewer 3 Responses:

Comment 1.)

“Firstly based on the title of the manuscript. The word LiDAR is used. To me this conjures up object detection and ranging. The manuscript seems more about imaging and not about object detection – certainly not ranging. Imaging and object detection are easy to relate, but ranging is a free space out-of-lab problem where the effective reflectivity can easily be lower than one part per billion. This is the most important quantum LiDAR problem and is not one that can easily be addressed by lab-based experiments. The authors do not seem to provide an overall detection probability (effective reflectivity) for a photon sent in to interrogate the target, which should be something like target intrinsic target reflectivity x collection efficiency x coupling losses x detector q.e. The authors should comment on the above, and/or provide a use-case for their (still impressive) results. If that use case is not actually some form of LiDAR they should consider changing to “quantum illumination” in the title.”

Response:

The reviewer brings up an important point. We agree that due to the issue of low effective reflectivity, referring to our protocol as a LiDAR is an overreach. We have changed our wording in the manuscript to reflect this fact. We do maintain that this protocol is motivated by LiDAR applications and that further improvements on our protocol can be made to make LiDAR applications viable. In this reply we answer first through what method we believe such improvements can be made. Second, why our group has the relevant expertise to pursue such a path. Third, what order of magnitude improvement must be made for application in the long term and shorter term.

As reviewer number two reasonably points out, effective reflectivity or probe channel loss poses a significant hurdle to Quantum LiDAR approaches and is not something we directly attempt to address in this work. Despite this, we believe our technique can, in principle, overcome this challenge while maintaining the signal to noise ratio advantages we reported. By multiplexing (spatially and spectrally) multiple modes from multiple sources one can increase the net probe flux and therefore be able to suffer more channel loss without increasing the pair-rate in any individual mode and losing the quantum SNR advantage.

Our group has substantial experience in designing, fabricating, and testing parametric down conversion sources (Kang, Anirban and Helmy) as well as embedding nonlinear sources such as frequency converters and parametric amplifiers in semiconductor diode lasers (Single-mode Bragg ring laser diodes) (Iu). These entangled photon pair sources may be electrically pumped and have the potential for spatial multiplexing at a large scale with the possibility thousands to tens of thousands of devices per chip. This has the potential to allow us to perform the same detection experiment thousands of times faster, addressing the problem of transmission loss.

To illustrate this point, we make an order of magnitude calculation on what photon flux might be needed to image a moving target. Consider a perfectly diffuse reflecting target moving at 100km/hr (27.28m/s). Given that a time bin can be on the order of 3ps, the spatial resolution is on the order of $3ps \cdot c = 0.8994mm$ where c is the speed of light. A target moving at 100km/hr will travel 0.8994mm in 32.378 μ s and so we must range this target within this time for accurate tracking. Supposing we use a similar pulse repetition rate of 80MHz the number of pulses that can be used to detect this target is $32378ns \cdot 80MHz = 2590.3$ pulses. Further, to achieve a reasonable SNR advantage of $\sim 20dB$ we can use a pair rate of 0.01pair/pulse giving an average of 25.90 photon pairs. Assuming a target reflectivity of around 10%, minimal atmospheric scattering, a range of $R = 100m$, a detector area of $1m^2$, and detector efficiency 80% this yields

$$\begin{aligned} P(R) &\approx P_0 \cdot \beta \cdot \frac{A}{R^2} \cdot \eta \\ &= 25.90 \cdot 0.1 \cdot \frac{1}{1 \times 10^4} \cdot 0.80 \\ &= 2.07 \times 10^{-3} \text{ counts} \end{aligned}$$

Where P_0 is the output LiDAR power, β is the reflectivity of the sample, A is the effective detector area, R is the range of the target, and η is the detector efficiency. This detection rate is much too low to for viable detection ranging. There are two approaches to overcoming this. First one may increase the pair rate. This naturally leads to a lower noise resilience however this may be a trade off one can afford. For example, sacrificing 10dB of noise resilience by increasing the pair rate by an order of magnitude would lead to an order of magnitude increased in receiver coincidence counts. Another approach is by increasing the experiment speed 10^4 times by using a 10^4 sources, we can expect ~ 20 coincidence counts before a target has moved enough to change the time-bin. This is large enough to enable tracking of the target. Thus, we believe spatial multiplexing used concomitantly with our coincidence counting regime can enable viable detection and ranging of moving targets and medium distances. Moreover, by measuring the time-bin we can range a target to less than 1mm precision.

The above LiDAR application is very ambitious and meant to show that some of the most demanding applications of LiDAR and target detection are not, in principle, out of reach. However, such applications are not in the immediate realm of possibility for our protocol. As such, we would also like to provide some short-term applications that require substantially less complexity. Biological imaging provides such an application because the time constraints in imaging a moving target are no longer present, and the range of the target is much smaller. In this context, the low power criterion is demanded not by clandestine operation but by sample damage and photobleaching. Noise sources include that of undesired autofluorescence.

Comment 2:

The same goes for the dispersion cancellation that is part of the main thrust of the manuscript. Could this reasonably help object identification outside the lab, in anything other than, say, a microscopy experiment?

Response:

To incorporate dispersion cancellation into an array-based system, more compact dispersive elements can be used. For example, a 2.7cm long chirped spiral Bragg grating waveguide (SBGW) with a slope of the linear dispersion of $-27.7ps/nm$ with a $0.3mm^2$ structure size and loss of 0.7dB/cm has been demonstrated (Sun, Wang and Deng). This is slightly lower than the dispersion used in our experiment ($90ps/nm$) however this is a relatively small reduction for the much smaller system footprint this can enable. With this, we believe there are approaches to dispersion cancellation which fit the scalability needs of a array based quantum temporal correlation based LiDAR.

Another consideration for implementing our protocol outside the lab is that the noise will not necessarily overlap with the probe photon temporally or spectrally. In order to address a concern raised by reviewer, we performed a proof-of-principal experiment using a modulator to chop the noise to overlap with the probe photon. We then scanned the delay of this modulation relative to the probe to locate the probe-reference peak location to show that ranging is, in principle, possible without apriori knowledge of the target location. We note that, due to time and laboratory limitations, this experiment is intended only to demonstrate the principal by which such a protocol could operate. To implement this technique effectively, further improvements are necessary and could be implemented given more time than exists to reply to the comments for this manuscript. We discuss which improvements are necessary at the end of the experimental results. The protocol and experimental results are as follows.

Protocol Description

We assume that the light sent to interrogate the target is pulsed with repetition rate τ_{rep} . This requirement can in fact be relaxed and a CW probe can be used in principle. To locate this peak initially it is not necessary to modulate the incoming signal at the photon signal duration ($\sim 20ps$ in our experiment based on the phase-matching bandwidth). It is only necessary to reduce the noise floor enough to effectively locate this peak. This process is illustrated in Fig 1 and is as follows:

1) Choose a duty cycle D (for example $\frac{1}{4}$ in Fig 1) and modulate the signal between pulses with a square pulse with this duty cycle.

2) Break up the period of repetition into subsections with duration $D \times \text{period}$.

3) Allow the non-local cancellation of dispersion to broaden the noise within the repetition rate.

4) Shift the phase of the pulse so that the unattenuated portion of the probe occupies the next section of the period (Fig1d).

5) Repeat until the peak is visible.

6) If the peak is not visible, the duty cycle can be further reduced.

7) Once the coincidence peak is located, the coincidence bin number can be used, in combination with a calibration coincidence measurement at a known distance to measure the range of the target.

Note also, once the peak is located, the duty cycle may be further reduced and one can 'lock on' to the coincidence peak.

Fig 1: Illustration of modulation scheme for peak location without apriori knowledge of target location. (a) Simulated raw coincidence data where peak is not visible. (b) Modulation of the first possible bin before dispersion. (c) Modulation of the first bin after dispersion of noise to be equal over the period. (d) Same as (a). (e) Modulation of the second bin before dispersion. (f) Modulation of the second bin after dispersion.

Experimental Description

To further probe the validity of this technique for peak location, we now document a proof-of-principal experiment implementing this search technique. The experimental setup is very similar to that of the manuscript and is depicted in Fig 2. The same Ti:Sapphire laser was used to pump the same PPLN ridge-waveguide, and the same WDMs were used to separate the probe and reference photons. The reference photon was directed directly towards a detector, with the inclusion of dispersion for one set of measurements. The probe is combined with CW noise originating from an LED whose spectrum was filtered using a Alnair Labs BVF-200CL Bandwidth-Variable Tunable Filter to match that of the probe photon as measured by the WDM (Supplementary Figure 6: $\sim 1560\text{-}1580\text{nm}$). The combined probe-noise path was directed towards a Lithium Niobate electro optic modulator. The modulator was controlled by a Highland Technology T-130 Electrical/ Optical Modulation Driver triggered by the electrical trigger signal from the Ti: Sapphire laser. The modulated probe-noise was then directed through, optionally, a dispersion shifted fiber towards the superconducting nanowire single-photon detector (SNSPD).

Due to current limitations on the available dispersion, modulation duty cycle, and loss from the modulation system as well as to reply in a timely fashion, we opted not to measure the range of a free space target. Instead, we prove that dispersion can be used to locate a coincidence peak in an environment plagued by CW noise strong enough to completely obscure the peak without the non-local dispersion cancellation technique. With more time, and better collection optics, this scheme could easily be extended to range a free-space target.

Fig 2: Experimental schematic for target ranging in a noisy environment.

Experimental Results

Due to the limited dispersion and modulation pulse width, we were only able to disperse the noise-reference coincidences by 2.26 times for a 1ns modulated noise pulse width. Thus, to observe improved peak distinguishability---and subsequent ranging---we used the lowest pulse width achievable (with reasonable jitter) from our modulator driver at 1ns allowing for the largest possible fractional peak broadening. The modulator delay was then swept from 0 to 12ns. For each 1ns delay increment, coincidence data was recorded for 30s. With more time and better equipment (such as the Berkely Nucleonics 765 800 MHz Fast Rise Time Pulse Generator which can achieve generate pulses with duration 300ps with ~10ps jitter), this recording time could be substantially reduced allowing for faster searching times.

We recorded histogram data from -40 to 40ns corresponding to 23.98m (Figure 3 (a) and (b)). For data processing we only examine the 6 coincidence peaks that remain within the window for all delays corresponding to 22.48m. We note however, that the true limitation here is imposed by the time-tagging device and is well beyond 80ns. The coincidence histogram with of 80ns was only for the purpose of visualizing the results and proof-of-principal.

Fig 4: (a) Coincidence histogram data for zero delay and 1ns modulated pulse width. (b) Coincidence histogram data with dispersion cancellation for zero delay and 1ns modulated pulse width.

The noise level was adjusted so that, in the absence of dispersion, the true coincidence peak was no longer distinguishable from noise. Precisely, using a 1ns window the classical SNR as defined in the main manuscript is $-30.73dB$ and the quantum non-dispersed SNR $-17.5dB$. The indistinguishability is shown in figure 4 (a) where the true coincidence peak takes on a value 1.77 time standard deviations above the mean.

For each delay, the coincidence peak height was calculated for each peak in the histogram and normalized by the mean peak height (Figure 4 (a) and (b)). Under the assumption that the noise is CW, a peak significantly above 1 indicates the location of the true coincidence peak. At a delay of 11.5ns pulse 3 takes on a value 3.58 standard deviations (one standard deviation 0.0116) above the mean (1). No such point exists in the non-dispersed data with the true coincidence peak (pulse 3 at 11.5ns delay) taking a value of 1.14. The standard deviation for the non-dispersed data is 0.0079.

Fig 4: (a) Normalized coincidence peak data for modulation delays 0-12ns and a modulation width of 1ns. (b) Normalized dispersed coincidence peak data for modulation delays 0-12ns and a modulation width of 1ns.

Using limited dispersion and limited modulation pulse width, we were able to recover a statistically significant coincidence peak from an environment plagued by noise where, without modulation and dispersion, the coincidence peak was indistinguishable. We believe this proof-of-principle experiment confirms that it is possible to detect and range a target's location without a priori knowledge of its range when the environmental noise is continuous. While these results are meant only to demonstrate the principle of this search technique, a more robust experimental setup could easily be developed given more time. Such a setup could significantly enhance the contrast of the true peak from the noise peaks after dispersion. The following are important considerations for such an apparatus:

- The amount of dispersion experienced by the probe/noise and reference paths must be increased without significant increase in loss. One of the main limitations that prevented us from adding more fiber to our setup was the mode mismatch between the anomalous dispersion fiber and the rest of the setup. Adding a second of such fibers would significantly increase the loss, prohibiting the demonstration of the protocol. A less lossy way of adding more dispersion to the setup would allow the system a much larger noise tolerance.
- To implement scanning and collection optics, the overall system loss must be reduced. This would involve redesign of the telescope, upgrading the various optical elements in the probe path, and the construction of a more permanent apparatus.
- The modulator driver's output pulse experienced a high jitter when the modulation pulse width was below 1ns. Smaller duty cycles will lead to a more significant dispersion and higher contrast between the true peak and noise peaks. A good piece of equipment for such purposes is the Berkely Nucleonics 765 800 MHz Fast Rise Time Pulse Generator which can achieve generate pulses with duration 300ps with ~10ps jitter.
- The SPDC source should also be tailored to the specific needs of the problem. The efficiency of our waveguide was not fully utilized as we were pumping a long waveguide with a femtosecond laser. A more optimal pair could lead to both higher flux, and a higher degree of time-frequency entanglement allowing for more optimal dispersion cancellation.

Comment 3:

On a much more minor note, there are also dispersion cancellation effects based on frequency-dependence of the loss that can change coincidence pulse widths nonlocally. Would this help their scheme or is the overall probe-arm loss frequency-independent?

Response:

The point raised by the reviewer is very interesting and an important consideration. Certainly, frequency dependent loss in the probe arm will affect the coincidence measurement results. In fact, since the

dispersion cancellation allows us to measure some amount of frequency correlation, and frequency dependent loss degrades the frequency correlation between the probe and reference, such loss will negatively impact our results. Fortunately, for many metals such as aluminum the reflectance is mostly constant over the wavelengths of interest 1400 – 1600nm. So, for imaging metal targets wavelength dependent loss is not a significant concern for practical implementations of our setup. It is worth noting that in order to avoid such frequency dependent loss, an achromatic collection optic must be designed to equally collect all frequencies of the probe light.

Comment 4:

The other main point is on limits. The claim is ~40dB improvement over a phase insensitive classical target detection scheme. I do not doubt it, but this will not be where the classical limit is. The best possible discrimination between the two possible detected states (target there or not) will be given by the Helstrom probability bound between them. The authors have shown that they are in a regime where they can easily beat *their* classical illumination, but not whether their improvement is better than classically allowed. It is likely that at the lowest probe powers in say fig 3b they beat the classical limit, but will be less clear for higher powers, because click detection is nonideal for the task of state discrimination. This point should be dealt with mostly in the supplementary material, but at least mentioned in the main text.

Response:

In this comment, the reviewer brings up several important points regarding our work in relation to the broader Quantum Illumination (QI) context. First, we will make explicit the problem which we are attempting to solve and why the optimal classical target detection lies outside the domain of possible solutions. Second, we will discuss the probe powers relevant to our problem scope and how our protocol behaves outside of this. Third, we will discuss how our protocol fits into the discussion of QI and what advantages we achieve in the language of quantum Chernoff bounds. We will compare this with Lloyds QI (Lloyd), as well as optimal classical target detection (Tan) to clarify explicitly where our advantage lies and where it does not.

The reviewer is indeed correct that click detection is not the theoretically optimum strategy for discrimination between the two possible detection scenarios (target present and target absent). This does mean our protocol is not better than classically allowed at discriminating between these two scenarios in a general context. However, our aim is not to develop a protocol that is better than classically allowed but to improve the noise resilience of current target detection techniques—such as LiDAR—where phase-sensitivity is untenable. This is motivated by current LiDAR implementations where phase-insensitive is often required due to the difficulty of phase stabilization resulting from target motion or substantial vibration. This phase-insensitivity excludes both optimal classical illumination and quantum illumination from the scope of our comparison. Further, as the reviewer suggests, the advantage afforded by our protocol does decrease with increasing probe power. For this reason, our protocol is aimed specifically at detection scenarios where high probe power is forbidden; for example, where clandestine operation is required. This limitation is shared with Quantum illumination (QI), where the average photon number per mode is required to be much less than one to see significant advantage.

These two conditions lead us to take as our ‘classical’ target detection protocol, that of single photon click detection as described by (Lloyd).

We would also like to point out that the 40dB improvement in SNR achieved in this work does not represent the largest possible improvement over our classical target detection. Using other non-local effects and leveraging other degrees of freedom we believe we can further increase this noise-resilience.

As the reviewer correctly suggests, our protocol is not better than classically allowed. We would, however, like to clarify, both in the broad context of QI and our narrower scope problem, what is meant by ‘better’ and how our work fits into the QI narrative. In Lloyd’s seminal work (Lloyd), he divides the state discrimination problem into two regimes; the ‘good’ regime where the signal power is greater than the noise power, and the ‘bad’ regime where the converse is true. There are then two ways one can talk about a ‘better’ protocol. First, one protocol may have a lower probability of error for a given number of trials than another protocol in the good or bad regimes. Second, the good regime of one protocol may extend further than the good regime of the other protocol. As we will explain in more detail, it is the latter case only in which our protocol beats Lloyds classical target detection. It was shown by (Tan) that a coherent state probe affords the lowest error probability of any classical-state (one with positive P representation) radar with the same transmitted energy. This detection protocol is realized with quantum limited heterodyne detection. For this protocol, there is no dependency on the transmissivity of the probe channel and so there is no ‘good’ or ‘bad’ regime and so this protocol is optimal in both senses. While we are trying to solve a problem that both optimal classical illumination (CI) and optimal QI cannot be applied to, we hope that presenting our protocol in a common language to that in which CI and QI are best discussed provides insight into how the benefits provided by our protocol fit into the broader QI context.

In Lloyd’s seminal work (Lloyd), he calculated the minimum probability of error and quantum Chernoff bound for two cases. The first, is for N-repeated transmissions of a single-photon pure state. The second, was with an optimum quantum measurement over N-repeated entangled-state transmissions. This gave the bounds for single-photon target detection

$$\Pr(e)_{SP} \leq \begin{cases} \frac{1}{2} e^{-\frac{N\kappa}{2}}, & \text{for } \kappa \ll 1, \text{ and } \kappa \gg N_b \ll MN_B \ll 1 \\ & \text{the 'good' regime.} \\ \frac{1}{2} e^{-\frac{N\kappa^2}{8N_b}}, & \text{for } \kappa \ll \frac{N_b}{M} \ll MN_b \ll 1 \text{ and } \kappa \gg \frac{N_B}{m} \\ & \text{the 'bad' regime.} \end{cases}$$

Where N is the number of single photons sent out, κ is the probe channel transmissivity, N_b is the background lights average photon number per temporal mode, and M is the number of temporal modes. For an entangled probe and reference photon the bound on the probability of error for optimal detection was found to be

$$\Pr(e)_{QI} \leq \begin{cases} \frac{1}{2} e^{-\frac{N\kappa}{2}}, & \text{for } \kappa \ll 1, MN_b \ll 1 \text{ and } \kappa \gg \frac{N_B}{M} \\ & \text{the 'good' regime.} \\ \frac{1}{2} e^{-\frac{N\kappa^2 M}{8N_b}}, & \text{for } \kappa \ll \frac{N_b}{M} \ll MN_b \ll 1 \text{ and } \kappa \gg \frac{N_B}{m} \\ & \text{the 'bad' regime.} \end{cases}$$

Both approaches have identical bounds in the 'good' regime however the 'good' regime for QI allows for M times higher background noise levels than single-photon detection. In the 'bad' regime, QI has an error exponent M times larger than single-photon detection meaning there is a lower probability of a false alarm.

It was later shown by Shapiro and Lloyd (Shapiro and Lloyd) that using a coherent state probe with non-zero expected photon number resulted in an quantum Chernoff bound,

$$\Pr(e)_{CS} \leq e^{\frac{1}{2}N\kappa(\sqrt{1-N_B}-\sqrt{N_B})^2}$$

Where N is the number of pulses sent to investigate the target. This inequality holds for all $0 \leq \kappa \leq 1$ and for all $N_b \geq 0$ and in the low background noise regime, reduces to

$$\Pr(e)_{CS} \leq e^{\frac{1}{2}N\kappa}$$

Which exactly matches that of Lloyd's QI. Implementation of such a coherent state detection protocol that achieve this bound must be phase sensitive and so are not within the bounds of the problem we are attempting to address.

The case of N -repeated transmissions of a single-photon pure state is analogous to our problem as in both cases, the goal is to distinguish two binomial distributions with different probabilities. In the case of single-photon target detection, the two binomial distributions are that of obtaining k single counts in N trials given the target is present, and that of obtaining k single counts in N trials given the target is absent. Similarly, in our case the first distribution is that of obtaining k coincidences in N trials given the target is present, and obtaining k coincidences in N trials given the target is absent. So the Chernoff bound for our DNCTD protocol is exactly that of Lloyd's single-photon illumination where κ is replaced by the probability of a true coincidence (P_T^D) and N_b is replaced by the probability of a false coincidence (P_{TF}^D) and $M = 1$. One necessary condition for this analogy is that coincidence counts are a sufficient statistic for target detection. That is, we gain no additional information by also examining the singles counts. This is equivalent to the assumption that the transmission of the reference photon is 100%. This is because, in this case, the number of true coincidences will be exactly equal to the number of probe singles (every returned probe photon results in exactly one coincidence).

The difference between our DNCTD scheme and Lloyd's single photon target detection is at what noise level N_b the discrimination problem enters the 'bad' regime. In Lloyd's case this crossing point is at $\kappa = N_b$. (Shapiro and Lloyd) Gives a reasonable estimate of M for Lloyd's QI at $M = 10^3$ modes. This increase the background noise level QI can remain in the 'good' regime by 30dB. Since the probability with which we detect a noise coincidence is reduced through coincidence counting and non-local cancellation of dispersion, we shift this 'good' to 'bad' crossing point ($P_T^D = P_F^D$) by the SNR ($\sim 40dB$).

References

- Iu, Meng Lon. "Electrically pumped efficient broadband cw frequency conversion in diode lasers using bulk χ^2 ." *APL Photonics* 5. 1 (2020): 011301.
- Kang, Dongpeng, Ankita Anirban and S Amr Helmy. "Monolithic semiconductor chips as a source for broadband wavelength multiplexed polarization entangled photons." *Optics express* 24.13 (2016): 15160-15170.
- Liu, Han, et al. "Enhancing LIDAR performance metrics using continuous-wave photon-pair sources." *Optica* 6.10 (2019): 1349-1355.
- Saikat, Guha and Baris I Erkmen. "Gaussian-state quantum illumination receivers for target detection." *Physical Review A* 80.5 (2009): 052310.
- "Single-mode Bragg ring laser diodes." *Optics Letters* 45.9 (2020): 2490-2493.
- Sun, Yu, et al. "Large Group Delay in Silicon-on-Insulator Chirped Spiral Bragg Grating Waveguide." *IEEE Photonics Journal* 13.5 (2021): 1-5.

REVIEWERS' COMMENTS

Reviewer #1 (Remarks to the Author):

The technical points I raised in the previous round of review have been satisfactorily addressed, and the additional results and clarifications made the importance of the work more convincing.

Reviewer #2 (Remarks to the Author):

I believe the authors have addressed all my concerns adequately and the additions/changes made, including those in reply to comments from other referees, have made the paper stronger. In my opinion, the manuscript is now suitable for publication.

Reviewer #3 (Remarks to the Author):

In response to my first comment the authors construct a detailed argument using physical values of parameters to show that with reasonable (detailed into reply) improvements it will become possible to range a moving target in a realistic time over a range of ~100m. The argument is fine but I would take issue with some of their estimated values. For example, for a non-specularly reflecting target the reflected light will on average be evenly spread over $2\pi - 4\pi$ steradians, which is of order 10. This factor should appear in the denominator of their equation, making the result nearer 2×10^{-4} counts.

Countering this is the fact that I believe that their tracking accuracy can and should be reduced. For an object at 100m a speed of 100km/hr is fast (~30m/s). Ranging at this distance for an object at this speed to within 1mm as suggested is not really required, for any object much larger than 1mm in size. Surely also at 100m the ranging beam itself will have a waist significantly larger than 1mm, so different parts of the beam will be reflected from bits of the target at different distances (unless the target is tiny), again suggesting that the tracking accuracy can be relaxed. This decrease in accuracy will increase the count rate, so overall I am happy with their conclusion.

The response of the authors to my second comment is neatly addressed by the experiment described in the reply. The experiment shows that the dispersion give significant improvement in the lab and I am convinced that some of this improvement will carry over into a more real world scenarios.

My third comment was more of a point of information for the authors and I am happy with their reply.

My fourth comment concerned classical bounds. I agree with the authors that the motivation to show orders of magnitude of improvement rather than beating what is allowable classically is perfectly reasonable. Beating the classical bounds at most intensities probably requires phase-sensitive detection. At optical frequencies this is practically useless for ranging outside the laboratory. Their detailed argument on bounding seems correct.

Overall the changes to the manuscript are appropriate and I support publication.

Reviewer 1 Replies

Comment 1.)

The technical points I raised in the previous round of review have been satisfactorily addressed, and the additional results and clarifications made the importance of the work more convincing.

Response:

Thank you very much for your contributions and feedback. We are very grateful for the insights you provided and the motivation for obtaining the additional results. We are happy to hear that you are satisfied with the additions to our manuscript.

Reviewer 2 Replies

Comment 1.)

I believe the authors have addressed all my concerns adequately and the additions/changes made, including those in reply to comments from other referees, have made the paper stronger. In my opinion, the manuscript is now suitable for publication.

Response:

We are happy to hear that your concerns have been addressed and we thank you for your invaluable insights. Your comments and advice have been very beneficial to us and helped us to write a stronger paper.

Reviewer 3 Replies

Comment 1.)

In response to my first comment the authors construct a detailed argument using physical values of parameters to show that with reasonable (detailed into reply) improvements it will become possible to range a moving target in a realistic time over a range of $\sim 100\text{m}$. The argument is fine but I would take issue with some of their estimated values. For example, for a non-specularly reflecting target the reflected light will on average be evenly spread over $2\pi - 4\pi$ steradians, which is of order 10. This factor should appear in the denominator of their equation, making the result nearer 2×10^{-4} counts. Countering this is the fact that I believe that their tracking accuracy can and should be reduced. For an object at 100m a speed of 100km/hr is fast ($\sim 30\text{m/s}$). Ranging at this distance for an object at this speed to within 1mm as suggested is not really required, for any object much larger than 1mm in size. Surely also at 100m the ranging beam itself will have a waist significantly larger than 1mm, so different parts of the beam will be reflected from bits of the target at different distances (unless the target is tiny), again

suggesting that the tracking accuracy can be relaxed. This decrease in accuracy will increase the count rate, so overall I am happy with their conclusion.

Response:

Thank you very much for this comment. This is a valuable addition to our calculation of what could, in-principal, be tracked and what kind of scaling would be required to achieve a viable LiDAR implementation. We are happy to hear you are satisfied with the overall argument and look forward to a future work where we can move closer to this limit.

Comment 2.)

The response of the authors to my second comment is neatly addressed by the experiment described in the reply. The experiment shows that the dispersion give significant improvement in the lab and I am convinced that some of this improvement will carry over into a more real world scenarios.

Response:

We are glad to hear that you found the additional experiment useful.

Comment 3.)

My third comment was more of a point of information for the authors and I am happy with their reply.

Response:

We are happy to hear you are satisfied with our reply.

Comment 4.)

My fourth comment concerned classical bounds. I agree with the authors that the motivation to show orders of magnitude of improvement rather than beating what is allowable classically is perfectly reasonable. Beating the classical bounds at most intensities probably requires phase-sensitive detection. At optical frequencies this is practically useless for ranging outside the laboratory. Their detailed argument on bounding seems correct.

Response:

We are glad that our argument proved convincing. Thank you very much for addressing this point as writing up the detailed bounding argument was an insightful exercise.

Comment 5.)

Overall the changes to the manuscript are appropriate and I support publication.

Response:

Thank you again for your insights and comments. They have greatly improved the strength of our work and we are happy to have had the opportunity to address them to your satisfaction.